# High Diversity of *Cryptosporidium* Species and Subtypes Identified in Cryptosporidiosis Acquired in Sweden and Abroad

**DOI:** 10.3390/pathogens10050523

**Published:** 2021-04-26

**Authors:** Marianne Lebbad, Jadwiga Winiecka-Krusnell, Christen Rune Stensvold, Jessica Beser

**Affiliations:** 1Department of Microbiology, Public Health Agency of Sweden, 171 82 Solna, Sweden; marianne.lebbad@outlook.com (M.L.); ilia.krusnell@gmail.com (J.W.-K.); 2Department of Bacteria, Parasites and Fungi, Statens Serum Institut, DK-2300 Copenhagen S, Denmark; run@ssi.dk

**Keywords:** molecular epidemiology, parasite, parasitology, epidemiology, genetic diversity, host specificity, Europe, Scandinavia, protist, sporozoa, zoonosis, zoonotic transmission

## Abstract

The intestinal protozoan parasite *Cryptosporidium* is an important cause of diarrheal disease worldwide. The aim of this study was to expand the knowledge on the molecular epidemiology of human cryptosporidiosis in Sweden to better understand transmission patterns and potential zoonotic sources. *Cryptosporidium*-positive fecal samples were collected between January 2013 and December 2014 from 12 regional clinical microbiology laboratories in Sweden. Species and subtype determination was achieved using small subunit ribosomal RNA and 60 kDa glycoprotein gene analysis. Samples were available for 398 patients, of whom 250 (63%) and 138 (35%) had acquired the infection in Sweden and abroad, respectively. Species identification was successful for 95% (379/398) of the samples, revealing 12 species/genotypes: *Cryptosporidium parvum* (*n* = 299), *C. hominis* (*n* = 49), *C. meleagridis* (*n* = 8), *C. cuniculus* (*n* = 5), *Cryptosporidium* chipmunk genotype I (*n* = 5), *C. felis* (*n* = 4), *C. erinacei* (*n* = 2), *C. ubiquitum* (*n* = 2), and one each of *C. suis*, *C. viatorum*, *C. ditrichi*, and *Cryptosporidium* horse genotype. One patient was co-infected with *C. parvum* and *C. hominis*. Subtyping was successful for all species/genotypes, except for *C. ditrichi*, and revealed large diversity, with 29 subtype families (including 4 novel ones: *C. parvum* IIr, IIs, IIt, and *Cryptosporidium* horse genotype VIc) and 81 different subtypes. The most common subtype families were IIa (*n* = 164) and IId (*n* = 118) for *C. parvum* and Ib (*n* = 26) and Ia (*n* = 12) for *C. hominis*. Infections caused by the zoonotic *C. parvum* subtype families IIa and IId dominated both in patients infected in Sweden and abroad, while most *C. hominis* cases were travel-related. Infections caused by non-*hominis* and non-*parvum* species were quite common (8%) and equally represented in cases infected in Sweden and abroad.

## 1. Introduction

Cryptosporidiosis is a global parasitic disease, which usually presents with self-limiting diarrhea. However, the disease can be severe, especially in immunocompromised and malnourished individuals [1]. The causative agent is the intestinal protozoan parasite *Cryptosporidium*, which infects a wide range of animals, including humans [2]. There are more than 40 recognized species described, but 2 in particular (*Cryptosporidium hominis* and *Cryptosporidium parvum*) account for most cases of human cryptosporidiosis [3]. In addition, around 20 other species/genotypes have also been observed in humans, including *Cryptosporidium meleagridis*, *Cryptosporidium cuniculus*, *Cryptosporidium ubiquitum*, *Cryptosporidium canis*, *Cryptosporidium felis*, *Cryptosporidium viatorum*, and *Cryptosporidium* chipmunk genotype I [3,4]. 

Cryptosporidiosis has been a notifiable disease in Sweden since 2004, with an increasing incidence from 69 (0.8 cases/100,000 inhabitants) cases annually in 2005 to 1088 (10.3/100,000 inhabitants) cases in 2019 [5]. There are several reasons for this increase, including better general knowledge of cryptosporidiosis, and the introduction of more sensitive diagnostic methods and, perhaps more importantly, multiple-agent diagnostic approaches, whereby clinicians do not need to specifically request testing for *Cryptosporidium.* Where this approach is used, the number of reported cases tends to increase [6,7]. Nevertheless, there is substantial local variation in the number of reported cases, which suggests that cryptosporidiosis is still underdiagnosed in Sweden [5]. 

Knowledge about the distribution of *Cryptosporidium* species and subtypes within a country is crucial for the management of cryptosporidiosis and to identify and understand transmission patterns and potential zoonotic sources [8]. The first larger molecular study of sporadic cryptosporidiosis cases in Sweden was conducted between April 2006 and November 2008. Patients from the Stockholm metropolitan area diagnosed with cryptosporidiosis were investigated, and 194 samples were successfully genotyped. A high occurrence of *C. parvum* was found; 111 cases vs. 65 of *C. hominis*. Only 17 (26%) of the *C. hominis* infections were acquired in Sweden, compared with 57 (51%) of the *C. parvum* infections. In addition, less common species such as *C. meleagridis* (*n* = 11), *C. felis* (*n* = 2), *C. viatorum* (*n* = 2), and *Cryptosporidium* chipmunk genotype I (*n* = 2) were found [9].

The largest known *Cryptosporidium* outbreaks in Sweden (and Europe) occurred in Östersund (Jämtland County) and Skellefteå (Västerbotten County) in 2010 and 2011. Both these outbreaks were caused by contamination of the municipal drinking water with *C. hominis* subtype IbA10G2, together affecting an estimated 45,500 persons [10,11]. *Cryptosporidium parvum* has been documented in water- and foodborne outbreaks as well as in outbreaks caused by animal contacts [12,13,14]. 

The primary aim of the present study was to identify if observations from the Stockholm metropolitan area (i.e., predominance of *C. parvum* but with relatively frequent detection of unusual species) reflected the situation in the rest of Sweden, and thereby to geographically expand the molecular investigation of cryptosporidiosis cases in Sweden. The second aim was to investigate if the *C. hominis* subtype IbA10G2 was established in the population following the large waterborne outbreaks in 2010 and 2011.

## 2. Materials and Methods

### 2.1. Invitation of Participating Laboratories

All clinical microbiology laboratories performing parasite diagnostic tests on human samples in Sweden were invited to participate in the study. The laboratories were asked to forward stool or fecal DNA from *Cryptosporidium*-positive cases to the Public Health Agency of Sweden (PHAS) for molecular species determination and subtyping, free of charge. Sample collection lasted for two years, from January 2013 to December 2014. All local departments of communicable disease control and prevention were also informed about the study, and the typing results were continuously submitted to the national mandatory notifications system (SmiNet).

### 2.2. Collection of Patient Data

Each submitted *Cryptosporidium*-positive sample was accompanied by information on the age, sex, and geographical location of the patient. Information concerning travel abroad within two weeks prior to onset of disease and, in some instances, assumed routes of transmission was retrieved from the referral and/or SmiNet and/or the local department of communicable disease control. 

### 2.3. Laboratory Investigations

The original *Cryptosporidium* diagnoses were made at local clinical laboratories using modified Ziehl–Neelsen staining or real-time PCR. In total, samples from 398 patients from 12 different microbiological laboratories were forwarded to PHAS for molecular analysis (Table 1). In total, 70 of the stool samples had been fixed in sodium acetate–acetic acid–formalin (SAF), while 328 samples consisted of stool without preservative and/or DNA extracted from unpreserved stool. Most of the SAF-fixed samples had been washed with phosphate buffered saline at the local laboratory before shipment. 

DNA was extracted directly from the stool specimens using a QIAamp DNA mini kit (Qiagen, Germany) according to the manufacturer’s recommendations. Prior to extraction, oocysts were disrupted using a bead-beater (Techtum, Sweden). DNA from external laboratories was extracted with local methods prior to submission to PHAS. To characterize *Cryptosporidium* species and subtypes, all samples were initially subjected to nested PCR of the small subunit rRNA (SSU rRNA) gene with subsequent restriction fragment length polymorphism (RFLP) and nested PCR of the 60 kDa glycoprotein (*gp60*) gene followed by sequencing [15,16,17]. In addition to RFLP analysis, bi-directional Sanger sequencing of the SSU rDNA amplicons was performed on (i) non-*hominis* and non-*parvum* isolates originally identified by RFLP, (ii) new or uncommon *gp60* subtype families, (iii) isolates where no amplification product was obtained at the *gp60* locus, and (iv) isolates where mixed species were suspected by RFLP. Species determination using a 70-kilo Dalton heat shock (*hsp70*) gene segment was performed on a limited number of samples where SSU rRNA PCR failed [18]. For the subtype determination of species not amplified with the Alves *gp60* primers [17], partial sequences of the *gp60* gene were amplified using different PCR protocols, depending on the investigated species, and products obtained by nested PCR were sequenced [19,20,21,22,23].

*Cryptosporidium*-specific actin and heat shock protein (*hsp70*) genes were amplified and sequenced in order to further characterize some uncommon species/genotypes [24,25]. 

Sequences were edited and analyzed using the BioEdit Sequence Alignment (version 7.0.9.0.) and compared with isolates in the GenBank database using the basic local alignment search tool (BLAST). All chromatograms were manually inspected for the presence of double peaks indicating mixed species/subtypes. In addition, *gp60* chromatograms and fasta files were examined using our in-house *gp60* sequence analyzer software CryptoTyper (unpublished).

Phylogenetic analysis was performed on SSU rDNA and *gp60* DNA sequences generated in the present study as well as sequences from known *Cryptosporidium* species and subtypes. Phylogenetic trees were generated using the neighbor-joining method based on Kimura’s 2-parameter model [26]. To estimate robustness, bootstrap proportions were computed after 1000 replications. Evolutionary analyses were conducted in MEGA X (https://www.megasoftware.net/ accessed on 24 April 2021).

Unique and uncommon nucleotide sequences were deposited in GenBank under the following accession numbers: *gp60*—KU727289, KU852701-KU852740; SSU rRNA—KU892559-KU892561, KU892564-KU892566; actin—KU892568, KU892571 and KU892572; *hsp70*—KU892574 and KU892577.

## 3. Results

### 3.1. Participating Laboratories and Patient Demographics

Samples were provided from 12 of the 21 clinical laboratories performing *Cryptosporidium* diagnostics in Sweden at the time of the study (representing 11 of the 21 counties) (Table 1). Altogether, samples from 398 patients (124 collected in 2013 and 274 in 2014) were received, representing 63% (398/628) of all cases reported in Sweden during these two years [5] (Figure 1). Among the participants, 217 (55%) were women and 181 (45%) were men. The age range was 1 to 88 years (mean (standard deviation), 34.7 (18.09) years); see Figure 1. The majority of the 398 isolates came from sporadic cases (*n* = 369), while 22 were related to outbreaks and seven to family clusters. 

### 3.2. Cryptosporidium Species Identified

Species identification was successful for 95% (379/398) of the samples, and 12 different species/genotypes were identified (Table 2). In total, 370 of the isolates were identified by RFLP and/or sequencing of the SSU rRNA gene, and 4 by sequencing of the *hsp70* gene (3 *C. parvum* and 1 *C. meleagridis*). For five samples with negative SSU rRNA PCR, species determination was based on the positive *gp60* result (4 *C. parvum* and 1 *C. hominis*). Of 19 samples that were PCR-negative and therefore not typeable (Table 1 and Table 2), 14 had been preserved in SAF. *Cryptosporidium parvum* was the species most commonly observed (79%; 299/379), followed in frequency by *C. hominis* (13%; 49/379), *C. meleagridis* (*n* = 8), *C. cuniculus* (*n* = 5), *Cryptosporidium* chipmunk genotype I (*n* = 5), *C. felis* (*n* = 4), *Cryptosporidium erinacei* (*n* = 2), *C. ubiquitum* (*n* = 2), and 1 each of *Cryptosporidium suis*, *C. viatorum*, *Cryptosporidium ditrichi*, and *Cryptosporidium* horse genotype. One patient was co-infected with *C. parvum* and *C. hominis*, as evidenced by RFLP analysis of the SSU RNA gene followed by Sanger sequencing of the same gene.

### 3.3. Origin of Infection

Out of the 398 patients, 250 (63%) were infected in Sweden, while 138 (35%) had acquired infection while traveling abroad to 55 different countries, representing all continents except Oceania and Antarctica, over the two weeks preceding disease onset. Ten cases had missing or uncertain data concerning origin of infection (Table 2). *Cryptosporidium parvum* was the most common species in cases of contracting infection in Sweden (84%; 211/250) and abroad (57%; 78/138); meanwhile, *C. hominis* was identified in 3% (8/250) of domestic cases and in 30% (41/138) of the cases infected abroad. 

*Cryptosporidium* chipmunk genotype I (*n* = 5), *C. ubiquitum* (*n* = 2), and *C. ditrichi* (*n* = 1) were found only in cases infected in Sweden, while *C. meleagridis* (*n* = 8), as well as *C. suis*, *C. viatorum*, and *Cryptosporidium* horse genotype (*n* = 1 each), were exclusively found in travel-related cases. *Cryptosporidium felis*, *C. erinacei*, and *C. cuniculus* were diagnosed both in domestic and travel-related cases (Table 2). 

### 3.4. Molecular Characterization of Cryptosporidium parvum

Subtyping using the *gp60* protocol was successful for 99% (296/300) of the *C. parvum*-positive samples, including the sample with mixed *C. hominis* and *C. parvum* (Table 3). In total, nine subtype families (IIa, IIc, IId, IIe, IIl, IIn, IIr, IIs, and IIt) and 42 subtypes were identified (Table 3). The most common subtype family was IIa, which was observed in 164 patients. It was represented by 22 different subtypes, of which IIaA16G1R1 (*n* = 42), IIaA15G2R1 (*n* = 31), and IIaA17G1R1 (*n* = 20) were the most common. The remaining 19 IIa subtypes were found in 1–13 patients each. Subtype family IId (*n* = 118 patients) was the second-most common subtype family, represented by 11 different subtypes, of which IIdA22G1 (*n* = 37) and IIdA20G1 (*n* = 24) were the most frequent. The remaining nine IId subtypes were found in 1–16 patients each. Three patients were co-infected with two *C. parvum* subtypes: one with IIaA14G2R1 and IIaA15G2R1, and two with IIaA15G2R1 and IIdA19G1 (Table 3). Subtypes from subtype families IIc, IIe, IIl, IIn, IIr, IIs, and IIt were found in one or two patients each (Table 3).

Among the IIa and IId subtypes identified in the present study, several sequence variations were observed, either in the conserved non-repetitive part or in the repetitive area. In Table 3 and Table 4, all subtypes and subtype variants are referred to via a corresponding GenBank acc. no. with 100% identity. For instance, in subtype IIaA16G1R1, the most common variant is IIaA16G1R1b (EU647727), which was found in 39 patients, and was the most common subtype in patients infected in Sweden (*n* = 30). Subtype IIaA16G1R1b_variant (KT895368), wherein the TCG repeat is located at a different position in the repeat region compared to IIaA16G1R1b, was found in two patients. This type of sequence variation has been described by Alsmark et al. [14], and it was found in four IIa subtypes, but not in any other subtype family (Table 3). Subtype variations were also seen in IId subtypes, exemplified by IIdA22G1, where three variants were identified, differing by one single nucleotide polymorphism (SNP) in the conserved part, AY166806 (*n* = 16), FJ917374 (IIdA22G1c; *n* = 19), and KR349103 (*n* = 2). Another type of sequence variation was found in IIaA14G1R1r1, which has an interruption of the TCA repeats by an ACA. This subtype, which was found in six patients, was named according to the proposed nomenclature by Amer et al. [28], who described a similar sequence variation, IIaA14G1R1r1b, in a sample from a calf. 

Three new *C. parvum gp60* subtype families, IIr, IIs and IIt, were identified in one patient each. The sequences showed 100% identity with *C. parvum* at the SSU rRNA gene: IIr and IIt to AF164102 (gene copy A), and IIs to LC270281 (gene copy B). At the *gp60* locus, subtype IItA13R1 (KU852718) was identified in a patient (Swec402) who had visited Tanzania prior to infection. The closest matches in GenBank were 93% to *C. parvum* subtype family IIb (AY166805) and 90% to IIp (MK956000). The other two patients (Swec447 and Swec434) were both infected in Sweden, the first with *gp60* subtype IIsA14G1 (KU852720), where the closest match was 95% identity to subtype family IIe (KU852716), and the second with subtype IIrA5G1 (KU852719), which showed 92% similarity to the *C. hominis* subtype family Ie (AY738184). This observation prompted us to investigate two additional genes, actin and *hsp70*. As regards the actin locus, the isolate was 100% identical to *C. parvum* sequence MH542350, and at the *hsp70* locus, where 1911 bp were successfully sequenced, it differed by eight SNPs compared with the IOWA *C. parvum* strain (XM625373) in the conserved part. These SNPs, however, did not display any changes in the amino acid sequence.

### 3.5. Molecular Characterization of Cryptosporidium hominis

*Gp60* subtyping was successful for 49 out of 50 *C. hominis* isolates (including the sample with mixed species) (Table 4). Seven subtype families, Ia, Ib, Id, Ie, If, Ii, and Ik, and 17 subtypes were identified. The most common subtype was IbA10G2 (*n* = 15), wherein 13 cases had contracted infection while abroad; the remaining 2 reflected domestic infections. The second-most common subtype was IbA9G3 (*n* = 9), detected in patients who were infected in nine different countries, mainly in Africa (Table 4). In addition to the two cases with IbA10G2, another three subtypes were found in patients infected with *C. hominis* in Sweden: IaA18R3 (*n* = 2), IfA12G1R5 (*n* = 1), and IkA18G1 (*n* = 2). The cases with the uncommon subtype families Ii and Ik have been described in a separate article [29].

### 3.6. Molecular Characterization of C. hominis/C. parvum Mixed Infection

One patient, a 4-year-old girl from Syria, had a mixed *C. hominis* and *C. parvum* infection with subtypes IaA18R3 and IIaA16R1.

### 3.7. Outbreaks and Family Clusters

Four outbreaks, all caused by *C. parvum*, were identified during the study period (Table 5). The first outbreak occurred among veterinary students in 2013, and two different subtypes were found, IIaA16G1R1 and IIdA24G1 [27]. Outbreaks 2, 3 and 4 were all foodborne and involved subtypes IIaA16G2R1, IIaA17R1, and IIdA17R1, respectively. Three small family clusters were also detected; two related to traveling, and one where a contaminated water well was the suspected vehicle (Table 5). 

### 3.8. Molecular Characterization of Additional Species 

Of 30 isolates from additional species, 28 were successfully sequenced at the SSU rRNA locus. Subtyping with the *gp60* protocol was successful for 29 isolates; the only exception was *C. ditrichi*, wherein no suitable primers were available (Table 6). 

A recently adopted child from Lithuania was infected with *C. suis*. A *gp60* sequence was achieved using the *Cryptosporidium* chipmunk primers [19], and by means of additional sequencing primers (data not shown) subtype XXVaR37 was obtained. This case and method will be described in a separate article.

Cases with *Cryptosporidium* chipmunk genotype I subtype XIVaA20G2T1 and *C. ubiquitum* subtype XIIa-1 were found in five and two patients, respectively, all infected in Sweden. 

Two patients were infected with *C. erinacei*. The SSU rDNA sequences from their samples were not identical; the sequence from patient Swec627 (KU892565) infected in Sweden differed by two SNPs from the closest *C. erinacei* match in GenBank (KC3056047), while the sequence from patient Swec653 infected in Greece was 100% identical to KC3056047 (Figure 2). The actin DNA sequence obtained from isolate Swec627 was 100% identical to a *C. erinacei* isolate (MN237648) from a European badger in Poland (unpublished). Based on *gp60* analysis, a new *C. erinacei* subtype, XIIIaA23R12, was identified in the first patient (Swec627), while subtype XIIIaA24R9 was found in the second patient (Swec653). 

Four patients were diagnosed with *C. felis*; three in Sweden, and one in Indonesia. The recently described *gp60* assay for *C. felis* [21] yielded four different sequence types. 

One patient infected in Kenya with *C. viatorum* subtype XVaA3b was identified during the study period. Material from this case was used to develop a *C. viatorum gp60* subtyping assay [22]. 

The eight patients diagnosed with *C. meleagridis* were all infected in Asia. Six of the samples were identified as genotype 1 at the SSU rRNA locus (AF112574), while sequencing of this gene failed for the remaining two isolates. Investigation of the *gp60* gene was successful for all eight isolates and revealed three subtype families, IIIb, IIIe and IIIg, and six different subtypes (Table 6). Subtype IIIbA23G1R1c (KU852727) differed by eight SNPs in the post-repetitive part of the sequence from IIIbA23G1R1b (KJ210609), and was named IIIbA23G1R1c according to the proposed nomenclature [23]. 

Two subtype families of *C. cuniculus*, Va and Vb, and five different *gp60* subtypes were observed. It was noted that the *C. cuniculus* Vb sequences published in GenBank carried 2–5 ACA repeats just after the TCA repeats (data not shown); thus, we followed the recommendation by Nolan et al. from 2010 [32] and included an R in the subtype designation, as in VbA29R4 for isolate Swe658, with 29 TCA repeats followed by 4 ACA repeats (KU852734). One patient infected in Greece carried a new subtype, VbA31R4. The sequence from patient Swec678 had the same numbers of TCA and ACA repeats as a VbA20R2 strain from the UK (GU971649), but differed by six SNPs and a 3 bp deletion in the post-repetitive part of the sequence, resulting in five amino acid changes. The subtype variant was designated VbA20R2b (KU852735), according to the proposed *gp60* nomenclature [33]. 

A third subtype family of *Cryptosporidium* horse genotype, VIc, was identified in a patient (Swe490) who had visited Kenya. This subtype, referred to as VIcA16 (KU852738), showed 88% and 87% identity to subtype families VIa and VIb, respectively. Four ACA repeats followed just after the TCA repeats, a feature also observed in *C. cuniculus* subtype family Vb. An extended molecular investigation including the SSU rRNA, actin, and *hsp70* genes was performed. The SSU rDNA sequence (KU892564) differed by one SNP from *Cryptosporidium* horse genotype sequences deposited in GenBank (FJ435962, MK775041). The closest match for the actin sequence (KU892571) was a sequence from *Cryptosporidium tyzzeri* (AF382343), from which it differed by eight SNPs. No actin sequences from the horse genotype were available in GenBank for comparison. The *hsp70* sequence KU892577 (1895 bp) showed 100% identity with the four *Cryptosporidium* horse genotype sequences available in GenBank. However, none of these sequences (298–403 bp) were long enough to cover the area with the 12 bp segment repeats towards the 3′ end of the *hsp70* gene. The horse genotype sequence from our study exhibited 10 repeats of a 12 bp segment with SNPs at the third and sixth bases—GG(C/T)GG(A/T)ATGCCA.

### 3.9. Phylogenetic Analyses 

A phylogenetic tree, which contained representative SSU rDNA sequences from all 12 species and genotypes detected in the present study and published sequences from most *Cryptosporidium* species/genotypes hitherto detected in humans, was constructed (Figure 2). In the *gp60* tree, one representative sequence from each subtype family detected in this study, except *C. felis* and *C. suis*, was included (*n* = 26). Sequences from established subtype families are clustered with sequences from this study. The new *C. parvum* subtype family IIt clustered with IIb, subtype family IIr with Ie, and subtype family IIs with IIe, while the new *Cryptosporidium* horse genotype subtype family VIc clustered with VIa and VIb (Figure 3).

## 4. Discussion

In the present study, performed between January 2013 and December 2014 and including samples from 398 patients with cryptosporidiosis, a high diversity of *Cryptosporidium* species and subtypes was identified. *Cryptosporidium parvum* was still the dominant species, and even fewer *C. hominis* cases were identified compared with a previous study performed in the Stockholm metropolitan area between April 2006 and November 2008. Meanwhile, the total number of patients infected with non-*hominis* and non-*parvum* species was quite high (8%), corroborating earlier observations [9].

Species determination was successful for 95% of the samples. *Cryptosporidium parvum* (79%) dominated both in patients infected in Sweden (84%) and abroad (57%), while *C. hominis* was much less common (13%) and identified in only 3% of the domestic cases and in 30% of the cases infected abroad. Mixed *C. hominis* and *C. parvum* infection was observed only in 1 patient, and for 10 of the patients infected with *C. parvum*, the origin of infection was unknown or uncertain. The high occurrence of *C. parvum* compared with *C. hominis* observed in Sweden is similar to the situation in other industrial countries, such as France, Ireland, and Canada [34,35,36], although a higher percentage of *C. hominis* has been observed in Spain and Australia [37,38]. Shifting trends have been seen over time in the Netherlands and New Zealand, showing the importance of longitudinal studies [39,40]. 

In terms of the analysis of cases according to age, the observed bimodal distribution (Figure 1) supports previous observations [9], with a relatively high number of cases observed among infants and toddlers and with a second peak—and the largest one—in the 30–44-year-olds. Quite similar age distributions have been observed in studies from Denmark, France, and Canada, but with the second peak in the 20–35-year-olds [34,36,41]. The bimodal age distribution may reflect transmission between parents and their children; however, no such family clusters were detected during the present study. There was a slight difference in the proportions of female and male cases: 55% and 45%, respectively. Interestingly, this difference in gender distribution has been seen in cryptosporidiosis cases in Sweden every year since 2005 (57% and 43% on average for women and men, respectively) [5]. A similar gender distribution was observed in neighboring Denmark in 2010–2014 [41]. Whether this difference in gender distribution reflects a higher awareness among females of the need to seek medical care, or something else, remains unknown.

A few *C. parvum* outbreaks were included in the present study (Table 5), but even if those patients (*n* = 22) are excluded from the dataset, *C. parvum* remains the dominant species in patients infected in Sweden, at least in the four areas that contributed most of the samples (Table 1). Differences in the geographical distribution of *C. parvum* and *C. hominis*, with more *C. parvum* in rural regions and more *C. hominis* in urban settings, have been described [42,43], and a similar tendency was seen in our study, where the rural region of Halland had relatively more cases of *C. parvum* and fewer cases of *C. hominis* compared with the metropolitan region of Stockholm (Table 1). These differences are often attributed to the closer contact with farm animals in rural areas and more traveling activity for people living in urban areas [44].

The subtyping of *C. parvum* showed a great variability; nine subtype families and 42 subtypes were observed. Two *C. parvum* subtype families, IIa (*n* = 164) and IId (*n* = 118), dominated both in patients infected in Sweden and abroad, while the other subtype families (IIc, IIe, IIl, IIn, IIr, IIs, IIt) were found only in one or two isolates each. The most common *C. parvum* subtype in the present study was IIaA16G1R1b (EU647727), observed in 39 patients. Detected in 24 sporadic cases and 4 cases related to an outbreak among veterinary students, it was also previously observed as the most common subtype amongst people infected in Sweden [9,27] (Table 5). This subtype is common in Swedish cattle and has been involved in earlier outbreaks and family clusters in Sweden [9,45], but was not reported amongst 48 subtyped *C. parvum* outbreaks between 2009 and 2017 in the UK [46]. 

Subtype IIaA15G2R1 is the dominating *C. parvum* subtype in many countries, both in cattle and in humans, and is responsible for numerous outbreaks of human cryptosporidiosis, probably due to its biological fitness and high virulence [41,46,47,48,49]. This subtype was found in 33 patients, of whom 16 were infected in Sweden; 13 were sporadic cases, and 3 cases were part of a family cluster (Table 5). This subtype has been detected in a few Swedish calves [50], but has not been associated with any known larger *Cryptosporidium* outbreak in Sweden. 

Many of the IIa and IId subtype sequences in the present study exhibited examples of polymorphism in the non-repetitive part, which is sometimes indicated by a lower-case letter when sequences are reported to GenBank [33], but this is not always the case. To overcome this dilemma, all subtypes and subtype variants in Table 3 and Table 4 are referred to via a specific GenBank acc. no. As an example, the most common IId subtype, IIdA22G1 (*n* = 37), occurred in three variants in this study, and one of them (AY166806) (*n* = 15) was only detected in domestic cases, most of them in connection to a foodborne outbreak, in which parsley was the suspected vehicle (Table 5). Another variant of this subtype, IIdA22G1c (FJ917374), was identified in 19 cases, of which 15 represented sporadic cases infected in Sweden. Both these variants of IIdA22G1 have been seen in earlier Swedish studies of cattle and humans [9,50]. The third variant of this subtype, KR349104, was seen in only two cases; one infected in France and one in Sweden. 

Subtype IIdA24G1 gained attention in 2011 in connection to a foodborne outbreak linking two Swedish cities, and it was also involved in an outbreak in 2013 among veterinary students occurring during the present study [12,27]. The same subtype, which is considered to be quite rare, was linked to 1 out of 48 *C. parvum* outbreaks in the UK where *gp60* subtyping was performed [46]. 

The subtypes mentioned above, IIdA22G1c and IIdA24G1, were involved in two foodborne outbreaks, which occurred simultaneously in late 2019 in 10 and 12 Swedish counties, respectively. Subtype IIdA22G1c was identified in 122 cases and IIdA24G1 in 86 cases. Spinach juice was the suspected vehicle for subtype IIdA22G1c, while no specific food item could be identified as a source of infection for IIdA24G1 [51].

Two new *C. parvum* subtypes were identified within the subtype families IIe and IIn. Subtype IIeA13G1 (KU852716) was found in a patient infected in Sweden, and subtype IInA10 (KU852717) in two patients; one infected in Tanzania, and one with unclear origin of infection (either Tanzania or Sweden (secondary infection)). *Cryptosporidium parvum* IIe is a well-known anthroponotic subtype family, but since only two cases (both humans from India) have been reported from subtype family IIn [52], it is impossible to say whether this one might be an anthroponotic or zoonotic subtype family. The same is true for the 3 new *C. parvum* subtype families, IIr, IIs and IIt, that were added to the 16 families already described [48]. Subtype IIrA5G1 and IIsA14G1 were found in patients infected in Sweden. Interestingly, a patient with subtype IIsA10G1 was recently described in a publication from Zambia [53]. The patient with subtype IItA13R1 was infected in Tanzania. 

One of the objectives of the present study was to investigate whether *C. hominis* subtype IbA10G2 had proliferated in Sweden after the large waterborne *C. hominis* outbreaks in Östersund and Skellefteå in 2010–2011, wherein an estimated 45,500 persons developed cryptosporidiosis. In the earlier study by Insulander et al. [9], performed from 2006 to 2008, 12 of the 17 *C. hominis* samples from patients infected in the Stockholm area carried subtype IbA10G2, while in the present study only two of the eight *C. hominis* patients infected in different parts of Sweden carried this subtype. The higher frequency of this subtype in the earlier study could be explained by the fact that several outbreaks at day care centers, with an index person infected abroad, appeared during that study period, while no such outbreaks were recorded in the current study. Few samples were submitted from the former outbreak areas; four from Jämtland County (Östersund) and four from Västerbotten County (Skellefteå), with all cases representing *C. parvum* (Table 1). The number of reported cases from these two counties during 2013 and 2014 was also low, at eight and five, respectively, and remained quite low during the following years up till now [5]. Therefore, even with these limitations in mind, we speculate that subtype IbA10G2 has failed to establish itself in Sweden following the large outbreaks. 

In the US, where IbA10G2 used to be the most common outbreak-related subtype, a new *C. hominis* subtype, IaA28R4, emerged in 2007 as a major subtype in sporadic cases and waterborne outbreaks [54]. Since 2013, subtype IfA12G1R5 has emerged as the dominant *C. hominis* subtype in the US, as well as in Western Australia [37,55]. In the present study, IaA28R4 was observed in two patients, both of whom had recently visited the US, and subtype IfA12G1R5 in two patients, one of whom was infected in Sweden and one in Germany.

*Cryptosporidium hominis* is generally regarded as an anthroponotic species only occasionally infecting other animals. However, recent studies have shown that for a certain subtype family, Ik, equines are natural hosts [56]. During the present study, the first human cases infected with this subtype family, Ik, were detected [29] (Table 4). Two unrelated patients, both infected in Sweden, carried the same *gp60* subtype, IkA18G1. This subtype was recently detected in a horse from China (MK770627)—information reinforcing the suspicion that zoonotic transmission might have occurred. Another uncommon *C. hominis* subtype family, Ii, was found in a father and son after returning from Thailand, where they visited a monkey farm. This *C. hominis* variant was previously described as *C. hominis* monkey genotype, and has rarely been detected in humans [29]. 

In the present study, 10 *Cryptosporidium* species/genotypes were found in addition to *C. parvum* and *C. hominis*, corresponding to 8% (30/379) of the total number of genotyped samples. This agrees with an earlier Swedish investigation performed on *Cryptosporidium* patients from the Stockholm metropolitan area, where 9% (17/194) of the genotyped samples represented species other than *C. hominis* and *C. parvum*. However, in the Stockholm study, the majority of these infections were acquired abroad; only three patients, two with *Cryptosporidium* chipmunk genotype I and one with *C. felis*, were infected in Sweden [9]. In the present study, an equal number of patients with species other than *C. hominis* or *C. parvum* were infected in Sweden (*n* = 15) and abroad (*n* = 15) (Table 1). This difference might reflect differences in study populations; in the first study, most patients originated from an urban area, while in the present study, many of the patients were from rural areas and probably more exposed to zoonotic *Cryptosporidium* species endemic to Sweden. 

The most common cause of non-*hominis* and non-*parvum* infections acquired in Sweden was *Cryptosporidium* chipmunk genotype I, which was diagnosed in five adults; four women and one man. Cryptosporidiosis caused by the chipmunk genotype is considered an emerging infection in the US [19,43]. Meanwhile, in Europe, only one human case (from France) has been reported outside Sweden [19]. The first Swedish cases were diagnosed in September 2007 and August 2008, at a time when no *gp60* subtype method was available for the chipmunk genotype I [57]. However, later analyses have shown that they carried the same subtype as the patients from the present study, XIVaA20G2T1, a subtype that recently was detected in red squirrels in Sweden (unpublished information). Red squirrels are consequently the most possible source of infection for our patients; the other described host animals—chipmunks, grey squirrels, and deer mice— are not native to the country [19,58]. Recent observations have shown a rising number of chipmunk genotype I infections acquired in Sweden, reflecting an emerging infection in this country (manuscript submitted). 

*Cryptosporidium erinacei* was first described in European hedgehogs, but has also been identified in other animals, such as horses and rats [59,60,61]. Most human cases have been reported from New Zealand (*n* = 13), while reports from Europe are scarce, including one case from the Czech Republic and one from France [40,62,63]. The European hedgehog is considered an endangered species in Sweden and other parts of Europe, which might explain the low number of reports from Europe compared with New Zealand, where it has become an invasive species since its introduction about 150 years ago. The two cases found during the present study are the first human *C. erinacei* cases reported from Sweden. The patient infected in Sweden carried a unique *gp60* subtype, XIIIaA23R12, while the patient infected in Greece carried subtype XIIIaA24R9, which has recently been reported in New Zealand [40]. 

*Cryptosporidium cuniculus* is a common species among rabbits worldwide, while human infection with this species has gained special attention in the UK, where, in addition to a documented outbreak, sporadic cases are quite common [64]. Reports of human infections from other parts of the world are increasing, and sporadic cases have been described in Nigeria, France, Spain, Australia, and New Zealand [40,48]. Two *C. cuniculus* subtype families have been described, Va and Vb. Four of the study patients were infected with Vb, which is the most commonly reported subtype family both in rabbits and humans, as well as in a few other animals, such as kangaroo and alpaca [65,66]. Subtype family Va, which was observed in one of the patients infected in Sweden, appears to be less frequent globally, and previous human infections were all reported in the UK [64]. These are the first documented human cases of *C. cuniculus* infection in Sweden.

One patient was infected with *C. viatorum* while visiting Kenya [22]. This species was initially thought to be anthroponotic because only human cases had been detected until 2018, when the first non-human host, an Australian swamp rat, was identified [67,68]. This swamp rat isolate was genetically quite different from all known human isolates, but recent studies from China focusing on different rat species have found *C. viatorum* isolates genetically similar to human isolates, and subtyping with *gp60* has demonstrated the same subtype, XVaA3g, in wild rats from China and a human from Australia [37,69].

Another visitor to Kenya was diagnosed with *Cryptosporidium* horse genotype. All horse and donkey samples positive for this species when sequencing of the *gp60* locus was performed carried subtype family VIa [70,71], which has also been found in one human case from Poland (MK784560) (unpublished). Only four cases with subtype family VIb have been described: two human cases, one in the UK and one in the US [54,72], and two cases in four-toed hedgehogs from Japan [73,74]. No reports of the VIb subtype family in equine hosts are available. The patient from our study carried a new *gp60* subtype family (VIc) of *Cryptosporidium* horse genotype. Since no other reports of this subtype family have been published to date, we cannot speculate on how this patient contracted the infection. 

Two children of 3 and 5 years of age, respectively, were infected in Sweden with *C. ubiquitum*, *gp60* subtype XIIa-1. These cases were unrelated, and no probable source of infection was identified. *Cryptosporidium ubiquitum* has a wide host range and has been found in wild and domestic ruminants, rodents, and primates, including humans. In the UK, it is believed that most human infections with *C. ubiquitum* are related to exposure to sheep, which are common carriers of subtype XIIa, while infected humans in the US mainly carry the rodent-associated subtypes XIIb and XIIc [20]. The only published *C. ubiquitum* case amongst Swedish animals is from a calf [45], and the occurrence in other animals in Sweden, including sheep, remains unknown and should be investigated.

The natural hosts of *C. suis* are domesticated pigs and wild boars. Human infections are not frequently diagnosed; only around 12 cases were reported prior to our case, which was a two 2-year-old adopted child diagnosed with cryptosporidiosis upon arrival in Sweden from Lithuania [48,75]. 

Human infection with the feline parasite *C. felis* is not uncommon; indeed, it is regarded as one of the five most common *Cryptosporidium* species infecting humans world-wide [48]. During the study period, four *C. felis* cases were detected, with three patients being infected in Sweden and one in Indonesia. Two of the Swedish patients had known contact with cats, and analyses of fecal samples from cats and their owners using a newly described *gp60* method showed that they shared *gp60* subtypes; thus, zoonotic infection was confirmed [21,30]. 

One study patient was infected with *C. ditrichi*, a recently described *Cryptosporidium* species in *Apodemus* spp. in Europe [76]. This was the first time *C. ditrichi* was diagnosed in a human, and later on, two more Swedish cases were identified [31]. 

Eight patients were infected with *C. meleagridis*, the third most common *Cryptosporidium* species infecting humans [77]. *Cryptosporidium meleagridis* has a wide host range, ranging from various birds to humans, and in some countries (e.g., Thailand and Peru), this species is reportedly more common in humans than *C. parvum* [77]. All patients infected with *C. meleagridis* in the present study had acquired infection while traveling in different Asian countries. The same *gp60* subtype was identified in two patients who had made the same journey to China, while the remaining six patients carried different subtypes (Table 6). Subtypes IIIeA17G2R1, IIIeA19G2R1 and IIIeA21G2R1 have been observed in chicken in China, and subtype IIIbA24G1R1 in poultry from Brazil, indicating zoonotic transmission as a possible route of infection [78,79]. Zoonotic transmission has been documented at a Swedish organic farm, where the infected person and the chickens carried the same *hsp70* and *gp60* subtypes [23,80]. 

When this study was initiated, the intention was to cover the entire country of Sweden, and all 21 clinical laboratories carrying out parasitology diagnostic tests in Sweden on a routine basis at the time of the study were invited to participate. One limitation is that by the end of the study period, only 12 of the regional laboratories (representing 11 different counties) had provided samples for the study. Our results, however, could very well be considered to reflect the reality, as nine counties only reported zero to one patient each with cryptosporidiosis during the study period (2013 and 2014) [5]. However, the number of referred samples varied considerably between the laboratories, with only four of them accounting for 95% of the submitted samples (Table 1). With this skewed distribution, we cannot conclude that the results of this study might be generalizable to the whole country. Another reason for this biased referral of samples could be the limited access to unfixed stool samples at some laboratories recommending that stool samples for parasitology be fixed in SAF. The intention at the beginning of the study was to avoid receiving samples fixed in SAF or formalin, which has a known negative effect on the PCR success rate [81], but in the second year, one laboratory with a high detection rate of *Cryptosporidium* and limited access to native (i.e., non-preserved) samples was invited to provide fixed specimens. In total, we received 70 samples fixed in SAF, 63 from this laboratory and 7 from other participating laboratories. The success rate of PCR for SAF-fixed samples was 80% (56/70), compared with 99% (323/328) for native stool samples. One recent study pointed out the limited availability of native stool material for the molecular analyses of stool parasites [34], but with the increased use of DNA-based diagnostics, the availability of unfixed material has increased, and the problem with fixed samples is in decline. 

Another potential bias is that the clinical laboratories performing the primary diagnostics used different testing strategies and detection methods. Some diagnostic real-time PCR methods are designed to primarily detect *C. parvum* and *C. hominis,* and do not target some of the less common and genetically distinct species, which might have led to an underrepresentation of these in some of the counties [82]. 

## 5. Conclusions

*Cryptosporidium parvum* was the dominant species both in cases infected abroad and in domestic cases. The observed occurrence of *C. hominis* was generally low, and no indication of an expansion of the subtype IbA10G2 previously causing the vast *Cryptosporidium* outbreaks in Sweden was found. There was a high diversity of species and subtypes, with 8% of the cases reflecting species other than *C. parvum* and *C. hominis*, some of which were found for the first time in humans in Sweden (e.g., *C. cuniculus, C. erinacei*, and *C. ubiquitum*). Our study also identified humans as a new host of *C. ditrichi*. Overall, zoonotic species and subtypes plays a major role in human cryptosporidiosis in Sweden.

## Figures and Tables

**Figure 1 pathogens-10-00523-f001:**
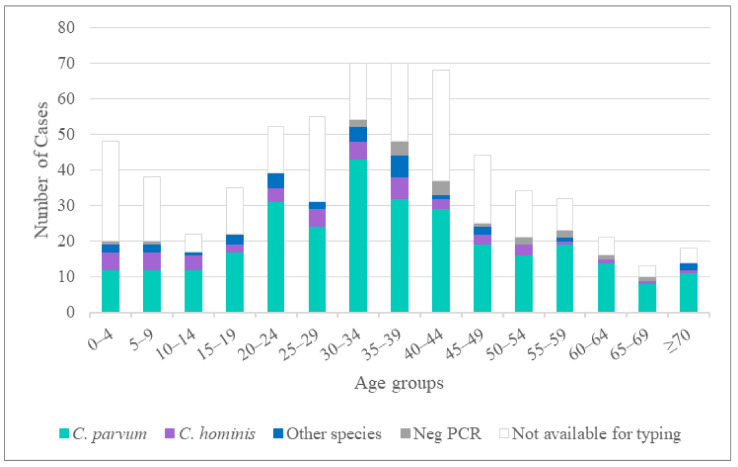
Number of reported cases of cryptosporidiosis detected in Sweden in 2013 and 2014 according to age group. One case with mixed *C. hominis* and *C. parvum* infection is not included in the figure.

**Figure 2 pathogens-10-00523-f002:**
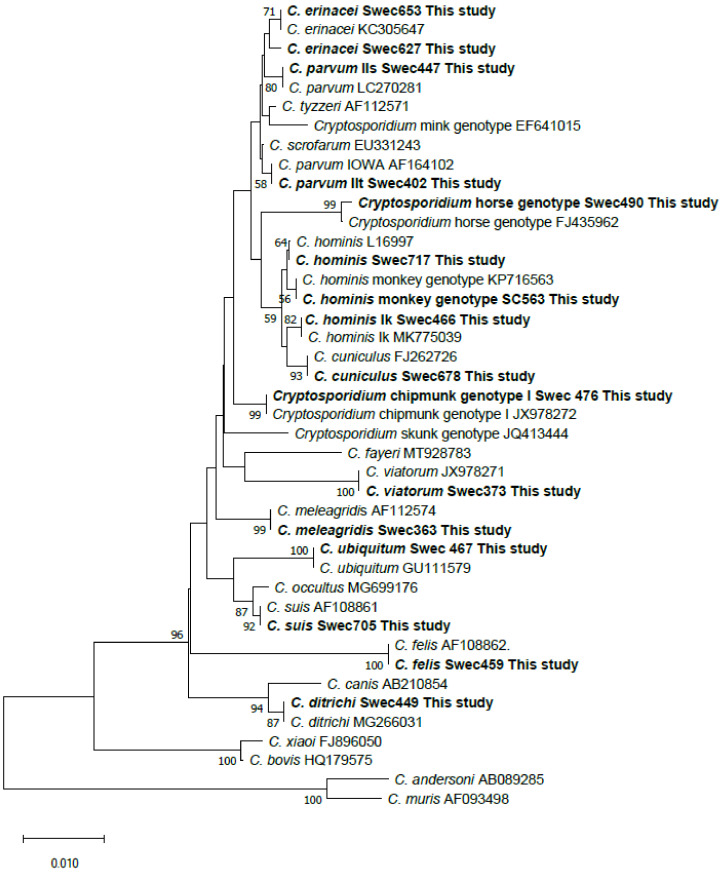
Phylogenetic relationships between partial SSU rDNA *Cryptosporidium* sequences obtained in the present study and sequences retrieved from the NCBI database. Trees were constructed using the neighbor-joining method based on genetic distance calculated by the Kimura’s 2-parameter model as implemented in MEGA X. The final dataset included 749 positions. Bootstrap values ≥50% from 1000 replicates are indicated at each node. Isolates from this study are indicated in boldface.

**Figure 3 pathogens-10-00523-f003:**
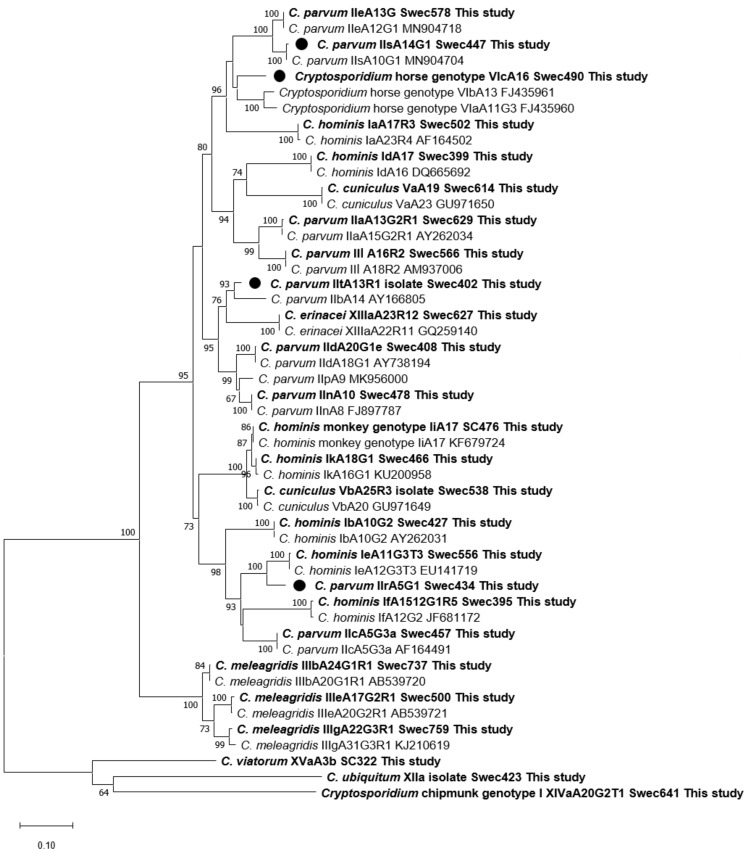
Phylogenetic relationships between partial *gp60 Cryptosporidium* sequences obtained in the present study and sequences retrieved from the NCBI database. Trees were constructed using the neighbor-joining method based on genetic distance calculated by the Kimura’s 2-parameter model as implemented in MEGA X. The final dataset included 840 positions. Bootstrap values ≥50% from 1000 replicates are indicated at each node. New subtype families observed in this study are indicated by filled circles. All isolates from this study are indicated in boldface.

**Table 1 pathogens-10-00523-t001:** *Cryptosporidium* species in samples provided by 12 participating laboratories across 11 Swedish counties and locally identified as *Cryptosporidium*-positive.

County	Number of Laboratories	Number of Samples	Species (Number of Samples)
Halland	1	138	*C. parvum* (112), *C. hominis* (15), *C. cuniculus* (5), *C. erinacei* (2), *C. meleagridis* (1), non-typeable (3)
Jämtland	1	4	*C. parvum* (4)
Jönköping	1	76	*C. parvum* (58), *C. hominis* (5), *Cryptosporidium* chipmunk genotype I (3), non-typeable (10)
Kronoberg	1	1	*C. felis* (1)
Skåne	1	5	*C. parvum* (3), *C. felis* (1), *Cryptosporidium* horse genotype (1)
Stockholm	2	92	*C. parvum* (62), *C. hominis* (23), *C. ubiquitum* (2), *C. ditrichi* (1), *C. felis* (1), *C. meleagridis* (1), *C. viatorum* (1), non-typeable (1)
Uppsala	1	70	*C. parvum* (52), *C. hominis* (6), *C. meleagridis* (6), *Cryptosporidium* chipmunk genotype I (2), *C. felis* (1), *C. hominis + C. parvum* (1), *C. suis* (1), non-typeable (1)
Västerbotten	1	4	*C. parvum* (4)
Västernorrland	1	2	*C. parvum* (2)
Västra Götaland	1	5	*C. parvum* (2), non-typeable (3)
Örebro	1	1	non-typeable (1)
Total	12	398	*C. parvum* (299), *C. hominis* (49), *C. parvum + C. hominis* (1), *C. meleagridis* (8), *C. cuniculus* (5), *Cryptosporidium* chipmunk genotype I (5), *C. felis* (4), *C. erinacei* (2), *C. ubiquitum* (2), *C. ditrichi* (1), *C. suis* (1), *C. viatorum* (1), *Cryptosporidium* horse genotype (1), non-typeable (19)

**Table 2 pathogens-10-00523-t002:** Distribution of *Cryptosporidium* spp. according to area of origin of infection among 398 patients with cryptosporidiosis diagnosed in Sweden from 2013 to 2014.

Species/Genotypes		Number of Samples (%)						
	Total	Sweden	Other European Countries	Africa	Asia	North America	South America	Unknown
*C. parvum*	299	211 (71)	60 (20)	9 (3)	5 (2)	3 (1)	1	10 (3)
*C. hominis*	49	8 (16)	10 (20)	14 (29)	13 (27)	3 (6)	1 (2)	0
*C. parvum + C. hominis*	1	0	0	0	1	0	0	0
*C. meleagridis*	8	0	0	0	8 (100)	0	0	0
*C. cuniculus*	5	3 (60)	2 (40)	0	0	0	0	0
*Cryptosporidium* chipmunk genotype I	5	5 (100)	0	0	0	0	0	0
*C. felis*	4	3 (75)	0	0	1 (25)	0	0	0
*C. erinacei*	2	1	1	0	0	0	0	0
*C. ubiquitum*	2	2	0	0	0	0	0	0
*C. suis*	1	0	1	0	0	0	0	0
*C. viatorum*	1	0	0	1	0	0	0	0
*Cryptosporidium* horse genotype	1	0	0	1	0	0	0	0
*C. ditrichi*	1	1	0	0	0	0	0	0
None-typable ^1^	19	16 (84)	1 (5)	2 (11)	0	0	0	0
All cases	398	250 (63)	75 (19)	27 (7)	28 (7)	6 (3)	2 (1)	10 (3)

^1^ Negative in all PCRs.

**Table 3 pathogens-10-00523-t003:** *Cryptosporidium parvum gp60* sequences generated in this study.

Species (No. of Patients)	Subtype Family (No. of Patients)	Subtype (No. of Patients)	GenBank Acc. No. ^1^	Origin of Infection (No. of Patients)
*C. parvum* (299)	IIa (164)	IIaA13G2R1 (2)	**KU852706**	Turkey (1)
			**KU852701**	Morocco (1)
		IIaA13R1 (1)	**KU852702**	Sweden (1)
		IIaA14G1R1 (13)	JQ030882	Cyprus (1) Georgia (2) Sweden (3)
			JX183798 (IIaA14G1R1b)	Sweden (7)
		IIaA14G1R1r1 (6)	**KU852703**	Sweden (4) Spain (1) Uzbekistan (1)
		IIaA14G2R1 (1)	KF128738	Greece (1)
		IIaA14R1 (2)	JX183797	Estonia (1) Sweden (1)
		IIaA15G1R1 (6)	AM937012	Sweden (4) unknown (1)
			**KU852704** (IIaA15G1R1_variant)	Sweden (1)
		IIaA15G2R1 (31)	AF164490	Dominican Republic (1) France (1) Mexico (1) Portugal (9) Portugal/Spain (1) Sweden (16) Venezuela (1) unknown (1)
		IIaA16G1R1 (42)	EU647727 (IIaA16G1R1b)	Georgia (1) Italy (1) Serbia (1) Spain (6) Sweden (30 ^2^)
			**KU852707**	unknown (1)
			KT895368 (IIaA16G1R1b_variant)	Austria (1) Sweden (1)
		IIaA16G2R1 (3)	DQ192505	Sweden (3)
		IIaA16G3R1 (2)	DQ192506	Portugal (1) Sweden (1)
		IIaA16R1 (1)	AM937010	Malta (1)
		IIaA17G1R1 (20)	GQ983359	Poland (1) Sweden (1)
			JX183801 (IIaA17G1R1c)	Italy (1) Sweden (11) South Africa (1)
			AF403168 (IIaA17G1R1c_variant)	Sweden (5)
		IIaA17R1 (1)	JX183800	Sweden (1)
		IIaA18G1R1 (12)	HQ005742	Finland (1)
			KF289038 (IIaA18G1R1b)	Sweden (2)
			KT895369 (IIaA18G1R1b_variant)	Portugal (1) Sweden (3)
			JX183803 (IIaA18G1R1d)	United Kingdom (1) Sweden (4)
		IIaA18R1 (1)	**KU852705**	Sweden (1)
		IIaA19G1R1 (3)	KC679056	Spain (1) Sweden (2)
		IIaA19G2R1 (1)	DQ630514	Mexico (1)
		IIaA20G1R1 (4)	KC995127	Croatia (1) Italy (1) Sweden (1) unknown (1)
		IIaA21G1R1 (5)	FJ917373	Sweden (5)
		IIaA22G1R1 (5)	JX183806	Sweden (4) unknown (1)
		IIaA23G1R1 (2)	KC995126	Sweden (2)
	IId (118)	IIdA16G1 (5)	FJ917372	Sweden (1)
			JX183808 (IIdA16G1b)	Spain (1) Sweden (3)
		IIdA17G1 (4)	**KU852708**	Italy (1) Norway (1) Spain (1) Sweden (1)
		IIdA18G1 (3)	AY738194	France (1)
			**KU852709**	Africa (1) Sweden (1)
		IIdA19G1 (16)	DQ280496	China (1) Morocco (2) Oman (1) Portugal (5) Sweden (2) Sweden/Portugal (1)
			**KU852711**	Sweden (3)
			**KU852713**	Sweden (1)
		IIdA20G1 (24)	AY738185 (IIdA20G1b)	Israel (1)
			JQ028866 (IIdA20G1e)	Sweden (20)
			**KU852711**	Croatia (1) Sweden (1)
			**KU852713**	Croatia (1)
		IIdA21G1 (3)	DQ280497	Portugal (1) Sweden (2)
		IIdA22G1 (37)	AY166806	Spain (1) Sweden (15)
			FJ917374 (IIdA22G1c)	Estonia (1) Greece (1) Sweden (15) Sweden/US (1) unknown (1)
			KR349103	France (1) Sweden (1)
		IIdA23G1 (5)	FJ917376	Ivory Coast (1) Sweden (4)
		IIdA24G1 (14)	JQ028865	Denmark (1) Germany (1) Sweden (11 ^3^)
			JX183810 (IIdA24G1c)	Sweden (1)
		IIdA25G1 (6)	JX043492	Sweden (6)
		IIdA29G1 (1)	GU458803	Sweden (1)
	IIc (2)	IIcA5G3a (1)	AF164491	Germany (1)
		IIcA5G3j (1)	HQ005749	Sweden (1)
	IIe (2)	IIeA10G1 (1)	KM539058	Guinea (1)
		IIeA13G1 (1)	**KU852716**	Sweden (1)
	IIl (1)	IIlA16R2 (1)	AM937007	Europe/Asia (1)
	IIn (2)	IInA10 (2)	**KU852717**	Tanzania/Sweden (1) Tanzania (1)
	IIr (1)	IIrA5G1 (1)	**KU852719**	Sweden (1)
	IIs (1)	IIsA10G1 (1)	**KU852720**	Sweden (1)
	IIt (1)	IItA13R1 (1)	**KU852718**	Tanzania (1)
Mixed subtypes	IIa + IIa (1)	IIaA14G2R1 + IIaA15G2R1 (1)	KF128738, KF128738	Italy (1)
Mixed subtypes	IIa + IId (2)	IIaA15G2R1 + IIdA19G1 (2)	AF164490, JF691561	Portugal (2)
*C. parvum* + *C. hominis* (1)	Ia + IIa (1)	IaA18R3 + IIaA16R1 (1)	KM538987, AM937010	Syria (1)
	neg *gp60* PCR (4)			Sweden (4)

^1^ Sequences submitted to GenBank during this study are indicated in boldface type. Accession numbers in non-boldface type refer to reference sequences from GenBank. ^2^ Four samples from an outbreak [27]; ^3^ two samples from an outbreak [27].

**Table 4 pathogens-10-00523-t004:** *Cryptosporidium hominis gp60* sequences generated in this study.

Species (No. of Patients)	Subtype Family (No. of Patients)	Subtype (No. of Patients)	GenBank Acc. No. ^1^	Origin of Infection (No. of Patients)
*C. hominis* (49)	Ia (12)	IaA17R3 (1)	**KU852723**	India (1)
		IaA18R3 (4)	JF927190	Sweden (2) Thailand (2)
		IaA18R4 (1)	FJ153246	Thailand (1)
		IaA20R3 (2)	**KU727289**	Tanzania (2)
		IaA23R3 (1)	JQ798143	India (1)
		IaA26R3 (1)	**KU852724**	Somalia (1)
		IaA28R4 (2)	KF682373	US (2)
	Ib (26)	IbA6G3 (1)	**KU852722**	Egypt (1)
		IbA9G3 (9)	DQ665688	Afghanistan (1) Ethiopia (1) Malawi (1) Mozambique (1) Somalia (1) Uzbekistan (1) Zambia (1)
			KF974523	Congo Republic (1) Uganda (1)
		IbA10G2 (15)	AY262031	Estonia (1) Greece (1) United Arab Emirates (2) Guatemala (1) Peru (1) Spain (7) Sweden (2)
		IbA13G3 (1)	KM539004	Burkina Faso (1)
	Id (3)	IdA16 (2)	HQ149034	China (1) Sri Lanka (1)
		IdA17 (1)	**KU852721**	Tanzania (1)
	Ie (1)	IeA11G3T3 (1)	GU214354	South Africa (1)
	If (2)	IfA12G1R5 (2)	HQ149036	Germany (1) Sweden (1)
	Ik (2)	IkA18G1 (2 ^2^)	KU727290	Sweden (2)
	Ii (2)	IiA17 (2 ^2^)	KF679724	Thailand (2)
	neg *gp60* PCR (1)			Sweden (1)

^1^ Sequences submitted to GenBank during this study are indicated in boldface type. Accession numbers in non-boldface type refer to reference sequences from GenBank. ^2^ Cases described in Lebbad et al., 2018 [29].

**Table 5 pathogens-10-00523-t005:** Outbreaks and family clusters involving *C. parvum* and *C. hominis* identified in the study period.

Outbreak/Cluster	Month/Year	Number of Suspected Cases	Number of Confirmed Cases	Number of Cases Typed	Information	Species	Subtype	GenBank Acc. No.
Outbreak 1 ^1^	Feb. 2013	13	6	6	Sweden: veterinary students	*C. parvum*	IIaA16G1R1 (4 cases) ^1^ IIdA24G1 (2 cases) ^1^	EU647727 (IIaA16G1R1b) JQ028865
Outbreak 2	Jan. 2013	10	2	2	Sweden: foodborne, private dinner, salad was the suspected source of infection	*C. parvum*	IIaA16G2R1	DQ192505
Outbreak 3	May 2014	8	2	1	Sweden: foodborne, private dinner, no identified source of infection	*C. parvum*	IIaA17R1	JX183800
Outbreak 4	March 2014	-	23	13	Sweden: foodborne, restaurant, parsley was the suspected source	*C. parvum*	IIdA22G1	AY166806
Cluster 1	July 2014	5	3	3	Sweden: a family, suspected contaminated water well	*C. parvum*	IIaA15G2R1	AF164490
Cluster 2	Nov. 2014	2	2	2	Portugal: a couple traveling together	*C. parvum*	IIaA15G2R1 + IIdA19G1	AF164490 + JF691561
Cluster 3 ^2^	Feb. 2013	3	2	2	Thailand: a father and his son traveling together ^2^	*C. hominis*	IiA17 ^2^	KF679724

^1^ Outbreak described in Kinross et al., 2015 [27]; ^2^ described in Lebbad et al., 2018 [29].

**Table 6 pathogens-10-00523-t006:** *Cryptosporidium gp60* sequences from non-*hominis* and non-*parvum Cryptosporidium* species generated in this study.

Species (No. of Patients)	Subtype Family (No. of Patients)	Subtype (No. of Patients)	GenBank Acc. No. ^1^	Origin of Infection (No of Patients)
*C. meleagridis* (8)	IIIb (3)	IIIbA23G1R1 (2)	KJ210606 (IIIbA23G1R1a) ^2^	Indonesia (1)
			**KU852727** (IIIbA23G1R1c)	Malaysia (1)
		IIIbA24G1R1 (1)	**KU852729**	China (1)
	IIIe (4)	IIIeA17G2R1 (2)	**KU852726**	China (2)
		IIIeA19G2R1 (1)	KJ210620 ^2^	Uzbekistan (1)
		IIIeA21G2R1 (1)	**KU852728**	Indonesia (1)
	IIIg (1)	IIIgA22G3R1 (1)	**KU852730**	Nepal (1)
*C. cuniculus* (5)	Va (1)	VaA19 (1)	**KU852733**	Sweden (1)
	Vb (4)	VbA20R2 (1)	**KU852735** (VbA20R2b)	Sweden (1)
		VbA25R3 (1)	**KU852731**	Spain (1)
		VbA29R4 (1)	**KU852734**	Sweden (1)
		VbA31R4 (1)	**KU852732**	Greece (1)
*C. erinacei* (2)	XIIIa (2)	XIIIaA23R12 (1)	**KU852736**	Sweden (1)
		XIIIaA24R9 (1)	**KU852737**	Greece (1)
*C. ubiquitum* (2)	XIIa (2)	XIIa-1 (2)	**KU852740**	Sweden (2)
*C. viatorum* (1)	XVa (1)	XVaA3b (1)	KP115937 ^3^	Kenya (1)
*Cryptosporidium* chipmunk genotype I (5)	XIVa (5)	XIVaA20G2T1 (5)	KP099089	Sweden (5)
*Cryptosporidium* horse genotype (1)	VIc (1)	VIcA16 (1)	**KU852738**	Kenya (1)
*C. felis* (4) ^4^	XIXa (3)	XIXa-39 (1)	MH240852 ^4^	Indonesia (1)
		XIXa-43 (1)	MH240856 ^4^	Sweden (1)
		XIXa-68 (1)	MH240883 ^4,5^	Sweden (1)
	XIXb (1)	XIXb-1 (1)	MH240901 ^4^	Sweden (1)
*C. suis* (1)	XXVa (1)	XXVaR37 (1)	MH187875	Lithuania (1)
*C. ditrichi* (1) ^6^	*gp60* PCR neg			Sweden (1)

^1^ Sequences submitted to GenBank during this study are indicated in boldface type. Accession numbers in non-boldface type refer to reference sequences from GenBank. ^2^ isolates included in Stensvold et al., 2014 [23]; ^3^ isolate included in Stensvold et al., 2015 [22]; ^4^ isolates included in Rojas et al., 2020 [21]; **^5^** case described in Beser et al., 2015 [30]; ^6^ case described in Beser et al., 2020 [31].

## Data Availability

Data presented in this study are available on request from the corresponding author, Jessica Beser. Due to existing general data protection rules, the data are not publicly available.

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
