# Peer review of "High Diversity of Cryptosporidium Species and Subtypes Identified in Cryptosporidiosis Acquired in Sweden and Abroad"

_pathogens, 2021, doi:10.3390/pathogens10050523_

Round 1
Reviewer 1 Report
Dear Authors,
I am pleased to read the content of interesting results of your publication. Congratulations on the work written clearly to the point. The paper is scientifically sound.
The experimental design is correct; the methodologies are correctly described and the results well discussed
I only have one editorial comment.
Line 107: please add a reference .
Author Response
Reviewer 2
Comments and Suggestions for Authors
Dear Authors,
I am pleased to read the content of interesting results of your publication. Congratulations on the work written clearly to the point. The paper is scientifically sound.
The experimental design is correct; the methodologies are correctly described and the results well discussed
I only have one editorial comment.
Line 107: please add a reference .
Response:
We thank the reviewer for noticing. References 15-17 have been added (line 107) – for some reason they had fallen out.
Reviewer 2 Report
The manuscript “High Diversity of Cryptosporidium Species and Sub-Types Identified in Cryptosporidiosis Acquired in Sweden and Abroad” presents data from a large and widespread study of Cryptosporidium species and subtypes in humans across a 2-year period in Sweden. The manuscript is well written, and I found the methodology appropriate and results clearly presented. The data presented in this manuscript are of interest and value. A few minor comments on wording are suggested below.
Line 24: How do you know they were zoonotic in origin?
Table 2: the title says country of infection, but only shows continent of origin if infection was associated with travel outside of Sweden. Consider rewording.
Line 474: line ends without punctuation and reads as if it was not fully edited.
Line 475: you say source, but you don’t know source. Consider rewording.
Author Response
Reviewer 2
Comments and Suggestions for Authors
The manuscript “High Diversity of Cryptosporidium Species and Sub-Types Identified in Cryptosporidiosis Acquired in Sweden and Abroad” presents data from a large and widespread study of Cryptosporidium species and subtypes in humans across a 2-year period in Sweden. The manuscript is well written, and I found the methodology appropriate and results clearly presented. The data presented in this manuscript are of interest and value. A few minor comments on wording are suggested below.
Line 24: How do you know they were zoonotic in origin?
Response:
The sentence has been reworded: infections caused by the zoonotic C. parvum subtype families IIa and IId dominated… (line 24)
Table 2: the title says country of infection, but only shows continent of origin if infection was associated with travel outside of Sweden. Consider rewording.
Response:
The title of Table 2 has been changed. Distribution of Cryptosporidium spp. according to area of origin of infection among 398 patients with cryptosporidiosis diagnosed in Sweden from 2013 to 2014
Line 474: line ends without punctuation and reads as if it was not fully edited.
Response:
We thank the reviewer for pointing this out. This has been corrected, and the sentence now ends like this: …endemic to Sweden (line 476)
Line 475: you say source, but you don’t know source. Consider rewording.
Response:
We agree with the reviewer and therefore changed the phrasing, as follows: The most common cause of non-hominis and non-parvum… (line 477)